# Influence of Herbal Medicines on HMGB1 Release, SARS-CoV-2 Viral Attachment, Acute Respiratory Failure, and Sepsis. A Literature Review

**DOI:** 10.3390/ijms21134639

**Published:** 2020-06-30

**Authors:** Marzena Wyganowska-Swiatkowska, Michal Nohawica, Katarzyna Grocholewicz, Gerard Nowak

**Affiliations:** 1Chair of Department of Dental Surgery and Periodontology, Poznan University of Medicinal Sciences, Bukowska 70, 60-812 Poznan, Poland; michal.nohawica1@gmail.com; 2Department of Interdisciplinary Dentistry, Pomeranian Medical University, Al. Powstancow Wlkp. 72, 70-111 Szczecin, Poland; kgrocholewicz@o2.pl; 3Department of Medicinal and Cosmetic Natural Products, Poznan University of Medicinal Sciences, Mazowiecka 33, 60-623 Poznan, Poland; gnowak.gerard@gmail.com

**Keywords:** herbal medicines, HMGB1, coronaviruses, SARS-CoV, respiratory system

## Abstract

By attaching to the angiotensin converting enzyme 2 (ACE2) protein on lung and intestinal cells, Sudden Acute Respiratory Syndrome (SARS-CoV-2) can cause respiratory and homeostatic difficulties leading to sepsis. The progression from acute respiratory failure to sepsis has been correlated with the release of high-mobility group box 1 protein (HMGB1). Lack of effective conventional treatment of this septic state has spiked an interest in alternative medicine. This review of herbal extracts has identified multiple candidates which can target the release of HMGB1 and potentially reduce mortality by preventing progression from respiratory distress to sepsis. Some of the identified mixtures have also been shown to interfere with viral attachment. Due to the wide variability in chemical superstructure of the components of assorted herbal extracts, common motifs have been identified. Looking at the most active compounds in each extract it becomes evident that as a group, phenolic compounds have a broad enzyme inhibiting function. They have been shown to act against the priming of SARS-CoV-2 attachment proteins by host and viral enzymes, and the release of HMGB1 by host immune cells. An argument for the value in a nonspecific inhibitory action has been drawn. Hopefully these findings can drive future drug development and clinical procedures.

## 1. Introduction

Viruses are one of the oldest organisms on Earth. They consist simply of a protein envelope and nucleic acids which renders them unable to replicate outside of a host [1]. Nevertheless, they are extremely adaptable and can rapidly alter their structure to suit the environment. While DNA viral copies are near exact, RNA viruses are more often dissimilar. Genetically compatible viruses can undergo recombination and create very difficult-to-treat infections [2].

By exchanging information, they can spread immunity against medical treatments, or the host immune system [3]. Orthomyxoviridae, an example of a family of RNA viruses which cause a disease commonly known as influenza or flu, can both rearrange compatible genes and mutate on a regular basis in order to remain invisible [4]. That is why a new flu vaccine is needed every year [5].

Due to quickly developing resistance to antiviral pharmaceuticals, vaccination is currently the most effective protection against viruses. However, there is building evidence that herbal medicines, which can be effective against viruses, do not develop resistance, help to stimulate the host immune system, and can also influence viruses in a specific way [6].

Coronaviruses are positive-stranded RNA viruses, allowing them to reproduce their proteins and genomes directly in the cytoplasm. Of all RNA viruses, they possess the largest genome [7]. This group of viruses, having a very high frequency of recombination, constantly produces many new variations. They can infect both birds and mammals [8].

In humans, they are mainly responsible for both mild respiratory infections [9], as well as the more severe Sudden Acute Respiratory Syndrome–SARS, Middle East Respiratory Syndrome–MERS [10] and Coronavirus Disease COVID-19 [11]. From the seven coronavirus species which infect humans that are currently identified, SARS-CoV-2 has the highest reproductive number [12].

Both SARS-CoV and the new SARS-CoV-2 are responsible for an acute respiratory failure. Other than the typical presentation of respiratory distress, about 25% of infected cases also develop acute digestive problems, suggesting that SARS-CoV-2 can spread through the oro-fecal route [13].

The SARS-CoV-2 infection can be divided in to three phases: binding to cells and incubation; virus release and cytokine storm; and a massive lymphocyte infiltration [14,15] associated with acute lung injury–these phases are similar to influenza [16].

However, their mechanisms are different. SARS viruses, instead of attaching to sialic acid links, bind directly to an integral membrane protein, angiotensin converting enzyme 2 (ACE-2), via the viral surface S protein [17,18]. Both SARS-CoV-2 S protein and influenza Haemagglutinin mediated cellular attachment mechanisms are dependent on priming, by proteases such as human airway trypsin-like protease (HAT) or transmembrane protease serine S1 member 2 (TMPRSS2), for subsequent entry. These human proteases recognize specific viral glycoproteins in order to facilitate their activation [19,20]. 

ACE-2 converts a vasoconstrictor (angiotensin II) into angiotensin (1-7), a vasodilator. SARS viruses attach to ACE-2 on the surface of lung, lymph and spleen epithelial cells. As a consequence of ACE-2 damage after a SARS viral infection, the renin-angiotensin system is down regulated, causing a decrease in oxygen saturation in lungs and pulmonary hemodynamics. [21,22]

Infected tracheobronchial and alveolar epithelial cells begin generating inflammatory cytokines and chemokines in response to the virus: interleukin-1 beta (IL-1β), IL-6, IL-18 (which causes spikes in interferon gamma (IFN-γ) production), C-C chemokine ligand 5 (CCL5, also known as RANTES), C-X-C chemokine ligand 10, and nuclear factor kappa-light-chain-enhancer of activated B cells (NF-κB). In a few hours following onset of infection, tumor necrosis factor alpha (TNF-α), IL-8, and monocyte chemoattractant protein-1 are secreted. These cytokines make the epithelial structures more porous, which allows for faster viral penetration, stimulates immune cell migration to the site of infection, and finally can lead to a cytokine storm and a more severe infection [23].

Studies investigating the effectiveness of immunosuppressive and anti-inflammatory drugs in ameliorating the damage caused by SARS-CoV-2 are already under way–combining their use with antibiotics [24].

However, while antibiotic therapy is commonly prescribed in sepsis to prevent a relapse due to nosocomial reinfection during resolution of organ damage [25], clinical immunosuppressive treatments for the SARS-CoV-2 associated acute respiratory failure and following sepsis have yielded as yet inconclusive results [26].

The limited success could be attributable to very high specificity of drugs tested, as the cytokine storm which they would try to inhibit involves multiple branches of the immune system across a wide range of signaling pathways, while broad range immunosuppression has been shown to worsen SARS-CoV-2 infection prognosis in transplant patients [27].

In response to high levels of NF-κB, TNF-α, RANTES, IL-6, and IFN-γ, macrophages and monocytes release high-mobility group box 1 protein (HMGB1). HMGB1 is stored and initially acetylated inside the nucleus. Upon immune cell stimulation with endotoxins such as bacterial endotoxin, lipopolysaccharide (LPS), or cytokines, HMGB1 becomes translocated to the cytoplasm and can actively undergo extracellular release through vesicles [28,29,30]. Alternatively, necrotic or virally infected non-immune cells can also release it passively [31,32,33,34]. HMGB1 can therefore also be released when the nuclei of cells are damaged. Its concentration becomes detectable around 8 h after stimulation [29,35]. Hypothetically, SARS-induced cytolysis, which occurs during the release of virus particles, could cause a rapid release of HMGB1 from infected cells [36]. Phylogenetically HMGB1 is well preserved. It shares 100% homology (in amino acid sequence) between mouse and rat, and a 99% homology between rodent and human genomes [37]. 

After release, HMGB1 binds to Receptor for Advanced Glycation End Products (RAGE). RAGE activates Mitogen-activated protein kinases and NF-κB, which in a positive feedback loop stimulate production of various cytokines [38,39]. This self-stimulating cytokine release causes a massive neutrophil infiltration into the lungs and subsequent acute lung injury. Acute respiratory failure, combined with a confirmed SARS-CoV-2 infection matches both old and new clinical sepsis criteria [40].

Common to the successful SARS-CoV-2 S protein attachment to ACE-2, attached S protein priming by TMPRSS2, and HMGB1 activation of the inflammatory sepsis, is the recognition of the substrate (S protein or HMGB1), by the target protease or receptor (TMPRSS2, ACE-2, RAGE) which is mediated by surface glycans [41,42]. This has been demonstrated with SARS-CoV S protein inhibition of cell entry via receptor glycan manipulation [43]. 

Herbal medicines can help to fight a viral infection in many ways. Most importantly they present an ability to block the SARS virus from binding to ACE-2 while simultaneously increasing ACE-2 activity and expression and have been shown to modulate the inflammatory cytokines levels–most notably IL-1 IL-1β and HMGB1. The adaptability of herbal medicines to various protein-inhibitory roles could be in part due to a shared glycan-mimicking structure of plant based phenolic compounds.

## 2. Interventions

### 2.1. Blocking the SARS Viruses Connection with ACE-2

SARS-CoV-2 entry has been successfully inhibited in vitro by direct ACE2 inhibition [5]. Targeting ACE2 would allow for specific intervention in the binding of Sars-CoV-2 [44]. Direct inhibition of ACE2 functioning could however have poor long-term effects on the body due to its importance in controlling blood pressure, as well as lung, kidney and cardiovascular function [45,46,47].

Pharmacological ACE inhibitors such as lisinopril are not effective in reducing ACE2 expression and at only 42% homology between the two enzymes, which can be inferred from their opposing physiological function, ACE inhibitors are not competitive enough at the ACE-2 attachment site with the SARS-CoV-2 to prevent infection [48].

Epidemiological studies assessing chronic conditions such as cardiovascular disease have found an association between dietary habits and prevalence [49,50]. Phenolic compounds from green tea, coffee and wine were identified as of specific preventative interest [51,52,53]. All phenol extracts from these foods were found to have a concentration dependent effect on inhibiting ACE activity; however, no data on ACE-2 was present [54].

Polyphenols, including green tea phenolic compounds, have a reputation for enzymatic inhibition, and have been reported to inhibit sugar digesting enzymes as well as lipases and proteases [55,56,57]. This has been noted well in the human digestive tract, which poses a barrier to their further phenolic absorption, but benefits from enzymatic inhibition in various, notably low-sugar, diets [58]. This can be attributed to phenolic compound dampening saccharolytic enzyme, e.g., invertase activity by 20% [59].

Crystal structure modeling of both ACE-2 and SARS-CoV-2 S protein has identified flavonoids, a subgroup of phenolic compounds, as suitable candidates for binding and inhibiting both molecules in silico [60]. Assessment has started on these compounds in vitro, and there are successes showing SARS-CoV-2 inhibition with phenolic compounds also effective against SARS-CoV [61,62].

Studies show that in order to complete the endocytic entry in to the cell, the SARS-CoV-2 virus needs further priming or cleavage of its S protein. This has been reported to be viable by TMPRSS2 [56,63] or cathepsin L [64] proteinases, with further studies supporting that their dual inhibition completely abolishes the coronavirus family MERS infectivity [65]. A broad spectrum, nontoxic protease inhibitor would be able to perform such a function. Further analysis of phenolic compounds could give more pharmacological guidance in this matter.

While long term proteinase and ACE-2 inhibition could be detrimental to cellular function and bodily homeostasis, targeted treatment partially reducing the effectiveness of coronavirus S protein attachment to the ACE2 or to the priming proteinase could have the potential to drop the SARS-CoV-2 viral load before a state of septic shock is reached at the peak of infection. 

### 2.2. HMGB1 Inhibition

HMGB1 might be a critical molecule to allow innate immune cells to respond to both infection and injury, and is necessary for mammalian survival as shown by a knockout murine model which displays lethal hypoglycemia [66].

Serum HMGB1 levels were consistently found to be significantly higher in septic patients who did not survive than those who did [35,67,68]. Intratracheal administration of HMGB1 has been shown to induce lung neutrophil infiltration, local production of proinflammatory cytokines (e.g., IL-1, and TNF), and acute lung injury [69].

Traditional medicine and hospitalization cannot offer much more against SARS-CoV-2 than symptomatic relief such as the use of oxygen. Some HMGB1 inhibitors like anti-IFN-γ antibodies, intravenous immunoglobulin, and minocycline seem to be helpful [70].

Anti-HMGB1 antibodies have also been shown to dose-dependently protect mice against lethal endotoxemia [35], as well as endotoxin-induced acute lung injury [69,71]

While normally corticosteroids reduce inflammation, they are helpless at fighting a HMGB1 induced cytokine storm [34]. Moreover, intravenous liquids containing nutrient solution and glucose may have serious side effects, such as significantly increased viral load or diabetic ketoacidosis [72].

On the other hand, insulin has a benefit of lowering HMGB1 levels [73]. Interestingly, both antithrombin III and thrombomodulin decrease HMGB1 in vitro [74].

A search of HMGB1 inhibitors shows a number of results, many of which are herbs and herbal constituents which have direct suppressive actions against this proteins activity. Counter intuitively, nicotine also significantly suppresses HMGB1 in the lungs [75].

### 2.3. Herbal Medicines

#### 2.3.1. *Angelica Sinensis* (Oliv.) Diels

The low molecular weight fraction of an aqueous extract of the Chinese herb *Angelica sinensis* (Dang Gui) is protective against lethal experimental sepsis and endotoxemia in a dose dependent manner. During laboratory induced lethal endotoxemia, 90% of mice which had the extract administered daily survived, vs. 30% survival in the control group, and during laboratory induced lethal sepsis 70% of mice survived when the extract was given daily, vs. 25% control. In the sepsis model, the administration of the extract did not begin until 24 h after the onset of sepsis, at which point some of the test subjects had already succumbed. This effect was in vitro shown to be non-cytotoxic to macrophages at any of the tested concentrations [76]. 

HMGB1 secretion was attenuated by the extract due to reduced translocation from the nucleus to the cytoplasm of activated macrophages, showing that even late administration of the Dang Gui extract significantly reduces serum levels of HMGB1, and can be attributed to the rescue of mice in part due to the attenuation of systemic HMGB1 accumulation. Inhibition of HMGB1 is therefore likely a key element in preventing septic shock induced death, as it is the only high mobility group protein with a cytokine-like activity which is important to starting and maintaining the septic response [77,78].

Ferulic acid and *A. sinensis* polysaccharide are the main components of Dang Gui extract, and the most biologically active. The polysaccharide has been extracted and shown to be inhibitory to viral replication in a murine leukemia virus in vivo model, also exhibiting anti-inflammatory properties [79]. Ferulic acid, while anti-inflammatory via inhibition of nitric oxide (NO) production, was excluded from *A. sinensis* mediated HMGB1 inhibition [76], however it does show some ability to inhibit human immunodeficiency virus (HIV) proteases [80].

#### 2.3.2. *Salvia Miltiorrhiza* Bunge

*Salvia miltiorrhiza* (Danshen) is also a natural remedy from the world of Chinese medicine which has proven experimentally to interact with HMGB1. Traditionally used to treat cardiovascular disorders, it was shown to be protective against lethal LPS-induced endotoxemia and sepsis by decreasing HMGB1 levels in vivo in a murine model [81]. Experiments with the Danshen extract (main bioactive ingredient: Danshensu) also demonstrated successful administration 24 h after the onset of sepsis. This proves that the HMGB1 inhibition by the herbal extract of Danshen can be preventatively inhibitory to the septic state, and also aid to arrest and reverse it. As an in vitro enterovirus treatment, ethyl acetate extracts of danshen were shown to have the strongest antiviral effect when administered along with the virus, suggesting interference in enterovirus entry mechanisms [82]. Danshen extract also possesses direct anti-viral activity as has been found to inhibit the Hepatitis B reverse transcriptase [83].

#### 2.3.3. *Camellia Sinensis* (L.) Kuntze (Green Tea)

Just as Dang Gui and Danshen, Epigallocatechin Gallate (EGCG–the main catechin found in green tea extracts) is an active natural extract able to rescue mice even if it is administered after the onset of induced sepsis. Camellia sinensis is a source of a polyphenolic group of compounds called catechins. The most prominent green tea catechins are EGCG, epigallocatechin, and epicatechin [84]. These molecules are well known for their antitumor, antioxidative, and antimicrobial activities [85,86].

Out of the three catechins found in green tea, only EGCG was found to inhibit LPS and cecal ligation and puncture (CLP) induced sepsis in a dose dependent manner, with up to 29% greater survival rate than control mice, even if treatment was delayed up to 24 h after the onset of sepsis. The effect was shown to coincide with EGCG (10 mM) almost completely eliminating HMGB1 release and macrophage cell surface clustering [87].

Green tea leaf (*Camellia sinensis* folium) extract also reduces endotoxin-induced release of HMGB1 and is therefore proposed to possess the ability to decrease mortality from sepsis if taken regularly. Complete inhibition of HMGB1 in vivo was seen under green tea extract doses as low as 10 μL/mL or 1 mL/kg, with no in vitro cytotoxicity of EGCG to macrophage cultures [88]. 

The mechanism through which EGCG interacts and modifies the kinetics of HMGB1 are still elusive. Reports show that EGCG can bind to lipid raft associated receptors [89,90,91,92]. Macrophages depend on lipid raft complexes to deliver LPS and signal an inflammatory state [93]. An analysis of structural motifs of EGCG could present a solution as to why it would locate to these lipid rafts, and whether its phenolic, enzyme inhibitory, activity plays a part in stopping the release of HMGB1. 

In addition, EGCG was shown to inhibit neuraminidase activity and viral genome synthesis of influenza, resulting in less effective cellular infection and viral replication. This disruption is thought to be caused by EGCG structural analogy to glycosidases which occupy cellular polysaccharides. This mimicry can inhibit Haemagglutinin and neuraminidase attachment to cellular membrane polysaccharide targets and stop influenza virus invasion and release respectively [94,95,96,97]. Haemagglutinin proteins, while not explicitly present in SARS-CoV-2, are present in the coronavirus family, and are utilized in cellular entry by the Human coronavirus HKU1 [98].

Other mechanisms have also been studied for different viral families. Hepatitis B viral (HBV) entry is dependent on Na+-taurocholate cotransporting polypeptide (NTCP) expressed at the membrane of human hepatocytes. A clathrin basket endocytosis, mediated by EGCG, caused a drop in NTCP surface expression and reduced HBV infectivity [99]. Herpes simplex virus (HSV) attachment was shown to be inhibited by the broad-spectrum activity of EGCG which successfully blocks attachment to heparan sulfate or sialic acid [100], while the speed of Ebola virus infection was slowed by EGCG inhibition of human cell-surface Heat shock protein A5 and therefore Ebola virus attachment [101,102].

#### 2.3.4. Panax Ginseng C.A.Mey.

Ginseng, rich in ginsenoside, is another herb with potent anti-inflammatory effects. The antiseptic activity of ginsenoside Rh1 (one of the main active constituents of the ginseng root extract) has been noted in HMGB1-activated human umbilical vein endothelial cells (HUVECs) and mice. Ginsenoside Rh1 increased the survival rate in a mouse sepsis model. It also significantly reduced HMGB1 release in LPS-activated HUVECs. Furthermore, it suppressed the production of TNF-α, IL-6, activation of NF-κB and extracellular signal-regulated kinase (ERK-1/2) by HMGB1. Ginsenoside Rh1 also inhibited HMGB1-mediated hyperpermeability and leukocyte migration in mice. In addition, treatment with ginsenoside Rh1 reduced the CLP-induced release of HMGB1, sepsis-related mortality and tissue injury in vivo [103]. 

An extract from Korean red ginseng significantly protected mice in experimental sepsis by decreasing TNF, IL-1, IL-6, and IFN-γ production via inhibition of NF-κB activation. It is likely that Korean ginseng will also reduce HMGB1 levels, because cytokines under the control of NF-kB, such as TNF and IFN-g, induce HMGB1 [104]. In fact, ginsenoside has been shown to decrease HMGB1 levels in a human uterine fibroid cell model [105]. Its direct anti-viral function has also been proven against Influenza strain H1N1, which was prevented from attaching to host α 2–3’ sialic acid receptors by ginsenoside interaction with viral Haemagglutinin when administered topically to the viral infection site intranasally [106]. 

#### 2.3.5. *Glycyrrhiza Glabra* L. (Licorice)

The main medicinal parts of licorice are its roots and rhizomes. Numerous studies have shown licorice to be antiviral against hepatitis C and HSV [107,108], anti-inflammatory [109,110], antioncogenic and antimicrobial [111].

Glycyrrhizic acid (GA) and glycyrrhetinic acid (GTA) are the specific chemical compounds that may be isolated from the licorice plant. It has been found that depletion of Sirtuin 6 (Sirt6) suppressed the number of human nasal epithelial cell cilia, and dramatically induced HMGB1 translocation from nucleus to cytoplasm in an epithelial tumor cell line. GTA has been shown to have anti-inflammatory and anti-allergic activity: Directly binding to HMGB1 protein extra-cellularly to inhibits its cytokine activities through a scavenger mechanism. In vitro studies using the 18-β-stereoisomer of GTA to enhance Sirt6 expression levels have shown inhibited translocation of HMGB1 protein from nucleus to cytoplasm and reversing its extracellular accumulation stimulated by LPS. [112]

GA was also found to significantly attenuate lung injury and decrease the production of inflammatory factors TNF-α, IL-1 IL-1β, and HMGB1, the release of which was stimulated with LPS treatment. GA also induces autophagy by enhancing the number of autophagosomes, possibly helping to deal with any necrotic tissue [113]. GA can efficiently block HMGB1 directly, and reduce its devastating effects [114].

#### 2.3.6. *Astragalus Mongholicus* Bunge

*Astragalus mongolicus* polysaccharide (APS) pretreatment has been shown to effectively inhibit HMGB1-induced increased permeability in lung endothelial cells (ECs). Signal transduction study has shown that APS inhibition of HMGB1 also affected a small guanylate Rho, and its downstream effector Rho kinase, in ECs, suggesting a multi layered involvement of APS [115].

#### 2.3.7. *Perilla Frutescens* (L.) Britton

Rosmarinic acid (RA) extracted from *Perilla frutescens* was shown to potently inhibit the release of HMGB1 and down-regulate HMGB1-dependent inflammatory responses in human endothelial cells. RA also inhibited HMGB1-mediated hyperpermeability and leukocyte migration in mice. Furthermore, RA reduced CLP-induced HMGB1 release and sepsis-related mortality [116]. RA has also proven effective in inhibiting viral replication and infection induced inflammation in a mouse model of Japanese encephalitis virus mediated Japanese Encephalitis [117].

#### 2.3.8. *Prunella Vulgaris* L.

The ethanol extract of *Prunella vulgaris* herb (EEPV) contains polysaccharides, flavonoids, and other phenols [118,119]. It inhibited HMGB1 release in LPS-activated macrophages in a PI3K-sensitive manner and reduced serum HMGB1 level and lung HMGB1 expression in cecal ligation and CLP-induced septic mice. EEPV is an inducer of Heme Oxygenase 1, which in turn reduces HMGB1 under LPS stimulus [120]. EEPV also has some antiseptic and anti-inflammatory potential [121]. It has been shown to exhibit anti-HIV, as well as anti-HSV (type 1 and 2) activity [119].

The Most active fraction of EEPV is rich in caffeic acid, hence termed caffeic acid-rich fraction [122]. *Prunella vulgaris* extracts have been proven to inhibit virus/cell interactions as well as host binding in HIV infection models [123].

#### 2.3.9. *Aspalathus Linearis* (Burm.f.) R. Dahlrgen (Rooibos)

Aspalathin and nothofagin extracted from Rooibos have been shown to effectively inhibit LPS-induced release of HMGB1, and suppressed HMGB1-mediated septic responses, such as hyperpermeability, adhesion and migration of leukocytes, and expression of cell adhesion molecules [124]. These molecules, as part of ethanol and alkaline extracts of the Rooibos plant, have also been shown to reduce Influenza A viral load at a late stage of the infection in vitro [125].

#### 2.3.10. *Cyclopia Intermedia* E. Mey. (Honeybush)

Vicenin-2 and scolymoside derived from Honeybush can also effectively inhibit LPS-induced release of HMGB1, and therefore suppress HMGB1-mediated septic responses such as hyperpermeability, the adhesion and migration of leukocytes, and the expression of cell adhesion molecules. In addition, vicenin-2 and scolymoside suppress the production of TNF-α and IL-6, and activation of NF-κB and ERK1/2 by HMGB1 [126].

#### 2.3.11. *Lonicera Caprifolium* L.

Intravenous treatment with *Lonicerae* flos, the main bioactive molecule of which is the chlorogenic acid, rescued LPS-intoxicated C57BL/6J mice under septic conditions and decreased the levels of cytokines such as TNF-α, IL-1β, and HMGB-1 in the blood [127,128].

#### 2.3.12. *Inula Helenium* L.

Extract of *Inulae* radix in LPS-activated RAW264.7 cells not only inhibited NF-κB luciferase activity, phosphorylation of IκBα, and iNOS/NO, COX-2/PGE2, HMGB1 release, but also significantly suppressed expression of intracellular and vascular adhesion molecules in TNF-α activated human umbilical vein endothelial cells [129]. The main active compound in *Inula helenium* is alantolactone, showing significant suppression of IL-8, TNF-α, and IL-1β release [130].

#### 2.3.13. *Rhodiola Rosea* L.

*Rhodiola radix* is a source of salidroside. The effect and mechanism of salidroside on sepsis-induced acute lung injury is mediated by the inhibition of inflammatory responses and HMGB1 production in bacterial LPS-treated macrophages and mice. Salidroside can also reverse the decreased Sirt1 protein expression in LPS-treated macrophages and mice [131]. Furthermore, salidroside was shown to alleviate the sepsis-induced lung edema, lipid peroxidation, and histopathological changes and mortality. Salidroside significantly decreases the serum TNF-α, IL-6, NO, and HMGB1 production, pulmonary inducible NO synthase and phosphorylated NF-κB-p65 protein expression, and pulmonary HMGB1 nuclear translocation in septic mice [132].

#### 2.3.14. *Abronia Nana* S. Watson

Boeravinone X, isolated from *Abronia nana*, has antiseptic effects. It was shown to inhibit LPS-induced release of HMGB1, and suppressed HMGB1-mediated septic responses, such as hyperpermeability, adhesion and migration of leukocytes, and expression of cell adhesion molecules. Comp 1 also suppressed the production of TNF-α and IL-6, and the activation of NF-κB and ERK1/2 by HMGB1 [133].

#### 2.3.15. *Cucurbitaceae* Sp.

*Cucurbitacin E* (CuE), a tetracyclic triterpene isolated from *Cucurbitaceae* plants, has proven to exert anti-inflammatory and immunologically regulatory activities. Recent findings highlight that CuE can ameliorate human bronchial epithelial cell insult and inflammation under LPS-stimulated asthmatic conditions by blocking the HMGB1-TLR4-NF-κB signaling [134].

#### 2.3.16. *Aconitum Carmichaelii* Debeaux

Use of the *Aconitum carmichaelii* tuber extract, of which the majority constituent is the C19-diterpenoid improved the liver function, decreased the pathological scores, and inhibited the expression of TLR4, NF-κB, HMGB1, and caspase-3 in a model rat liver injury treatment [135,136].

#### 2.3.17. *Plumbago Zeylanica*

Plumbagin prominently hampered HMGB1 expression and subsequently quelled inflammatory cascades, as NF-κB, TNF-α and myeloperoxidase activity. It also interrupted the reactive oxygen species-HMGB1 loop as evident by restored liver function [137].

#### 2.3.18. *Ecklonia Cava*

Brown algae have been recognized as a food ingredient and health food supplement in Japan and Korea, and phlorotannins are unique marine phenol compounds produced exclusively by brown algae. Phlorotannin rich extracts of the edible brown alga *Ecklonia cava* were investigated against the hyper-inflammatory response in LPS-induced septic shock mouse model. *E. cava* extract significantly increased the survival rate and attenuated liver and kidney damage in mice. In addition, *E. cava* attenuated serum levels of NO, PGE2, and HMGB-1. In macrophages, treatment with *E. cava* extract down-regulated iNOS, COX-2, TNF-α, IL-6, and HMGB-1. In addition, *E. cava* suppressed the NIK/TAK1/IKK/IκB/NFκB pathway. Dieckol, a major compound in the extract, reduced mortality, tissue toxicity, and serum levels of the inflammatory factors in septic mice [138]. In terms of SARS viral infectivity, Dieckol has been shown to inhibit the SARS-CoV 3CLpro cysteine protease, therefore inhibiting the viral ability to replicate [139]. SARS-CoV 3CLpro shares 99.02% sequence identity with SARS-CoV-2 3CLpro [140].

#### 2.3.19. *Actinidia Argute* (Siebold and Zucc) Planch ex Miq.

Kiwifruit peel extract is rich in polyphenols, the main of which are procyanidins representing 92% *w*/*w* of the total. A kiwifruit extract inhibited the production of inflammatory molecules such as IL-6, IL-1β, TNF-α pro-inflammatory cytokines, HMGB1 and granzyme B serine protease by activated monocytes [141].

### 2.4. Active Natural Compounds

#### 2.4.1. Pelargonidin

Study has shown that pelargonidin (PEL) had effectively inhibited LPS-induced release of HMGB1 and suppressed HMGB1-mediated septic responses, such as hyperpermeability, adhesion and migration of leukocytes, and expression of cell adhesion molecules. Furthermore, PEL inhibited the HMGB1-mediated production of TNF-α and IL-6, as well as NF-κB and ERK1/2 activation [142].

#### 2.4.2. Flavonoides

Luteolin was demonstrated to reduce the release of HMGB1 through destabilizing c-Jun and suppressing HMGB1-induced aggravation of inflammatory cascade through reducing Akt protein levels [143].

Quercetin reduced the lung permeability changes and neutrophil and macrophage recruitment to the bronchoalveolar fluid compared to the placebo mouse model. Additionally, quercetin significantly reduced COX-2, HMGB1, iNOS expression, and NF-κB p65 phosphorylation. These results suggest that Quercetin may be a promising potential therapeutic agent against sepsis [144].

Baicalein from root of *Scutellaria baicalensis* also significantly down-regulated the expression of Matrix Metalloproteinase-2/9 and attenuated HMGB1 translocation from the nucleus to the cytoplasm [145].

Quercetin, Luteolin, and Baicalein have all been found to inhibit the SARS-CoV 3CLpro protease [146,147]. Quercetin has also been identified to target the same enzyme in MERS-CoV, as it has a flavonoid structure with carbohydrate attachments which favor localization to the S1 and S2 sites on the S protein [148].

Resveratrol has also been shown to inhibit HMGB1. HMGB1 migrates out of the nucleus during Dengue Virus infection. This migration is inhibited by resveratrol treatment and is mediated by induction of Sirt1 which leads to the retention of HMGB1 in the nucleus. The enhanced transcription of interferon-stimulated genes by nuclear HMGB1 also contributes to the antiviral activity of resveratrol against dengue virus [149].

## 3. Discussion

In murine and human serum of septic subjects HMGB1 persists to be secreted for a long time. Peak levels in cell cultures are only reached 18 h after stimulation [35]. In contrast to other cytokines such as TNF with a peak at 90 min from initial stimulus, HMGB1 generates further waves of inflammatory cytokine production through RAGE, and toll-like receptors 2 and 4 [150,151,152,153]. HMGB1 and other inflammatory cytokines are persistently elevated in sepsis [154,155].

All of the identified herbal extracts and flavonoids exert suppression of HMGB1 activity. This is often accompanied by a drop in NF-κB activation, and reduction of TNF-α, IL-1, IL-6, and IFN-γ, signifying a reduction in the inflammatory response. Multiple compounds boast anti-proteinase activity. Even when the extract’s anti-inflammatory effect is not direct, in multiple cases there are indications that it is due to an inhibition of a surface protease, which is either an effector of the internal inflammatory changes or is in turn necessary to cause them.

The anti-proteolytic activity of herbal extracts covers an anti-inflammatory, as well as anti-viral effect across a variety of different viruses.

Many herbal extracts contain large groups of closely related polyphenols, making it difficult to separate them, and even more difficult to attribute the effect of the extract to just one of them. However, there is a lot of similarity in the effects of all the different extracts, just as there is similarity in the chemical structure of their main constituent molecules Table 1.

## 4. Conclusions

The pathogenesis of sepsis is attributable, at least in part, to dysregulated systemic inflammatory responses characterized by excessive accumulation of various proinflammatory cytokines. A ubiquitous nuclear protein HMGB1 is released by activated macrophages/monocytes in late stages of SARS-CoV-2 infection, and functions as a late mediator of lethal endotoxaemia and sepsis. Circulating HMGB1 levels are elevated in a delayed fashion (after 16–32 h) in endotoxaemic and septic animals. Administration of recombinant HMGB1 to mice recapitulates many clinical signs of sepsis, including fever, derangement of intestinal barrier function, lung injury, and lethal multiple organ failure. Anti-HMGB1 antibodies or inhibitors (e.g., ethyl pyruvate, nicotine, stearoyl lysophosphatidylcholine and Chinese herbs such as *Angelica sinensis*) protects mice against lethal endotoxaemia, and rescues them even when the first doses are given 24 h after onset of sepsis.

Most of the active compounds from the selected herbal medicines discussed here have aromatic ring structures, some studded with hydroxyl groups, almost all having pentose or hexose sugars, and occasionally nitro, sulphate or acetoxy functional groups as illustrated in Table 1. Experimental data suggests the majority have the ability to interrupt HMGB1 release and function, while some have also been shown to directly interrupt viral attachment and release. Data regarding the exact binding moieties of herbal extracts is, however, still missing.

It is unclear whether these functions are possible due to their specific chemical configuration which targets only viral or host proteins responsible for attachment, invasion, replication or release of the virus particle, or weather instead their general ability to inhibit a variety of glyosidic processes. Effective enzymatic inhibition, competitive or not, could be attributed to a couple of factors. These molecules are amphiphilic allowing them to effectively embed within lipid membranes and protein rafts. Common among the molecules is a multi-benzene ring steroid-like adaptation which serves as the lipophilic element, while a close resemblance of motifs to human glycolipids and glycoproteins can serve as a substrate for human and viral proteinases. This mimicry could slow down a variety of processes, hypothetically even inhibiting a virus which is adapted to deal with host specific glycoproteins and lipids, while ill prepared to digest plant analogues.

Generally, phenols are capable of disrupting multiple enzymatic processes, both human and viral. As the human body is still the most effective weapon against infection, it is paramount to give the immune system enough time to mount an attack. A diffuse inhibition of viral and cellular surface proteolytic processes could limit the speed of viral infection and cellular damage, giving the immune system some extra time to fight.

Experimental data established HMGB1 as a late mediator of lethal endotoxemia and sepsis, with a wide–over 24 h–therapeutic window, which allows for the clinical management of lethal systemic inflammatory diseases. Multiple therapeutic agents from herbal medicines have been identified as candidate compounds for the task. An analysis of combinations of extracts with complimentary functions, or a pharmacological generation of an optimum molecule, could further this field. Molecular studies on protease inhibitors with human-polysaccharide or viral-polysaccharide mimicking docking motifs and difficult to digest phenolic groups could prove to be the most effective remedy against a well-established viral infection. Clearer proof of direct inhibition of target proteases, molecules such as HMGB1, and receptors would provide a strong basis for pharmacological development. With many animal model studies already establishing a clear link in sepsis attenuation by HMGB1 inhibition, it is now time to assess if these findings can translate on to human models.

## Figures and Tables

**Table 1 ijms-21-04639-t001:** Chemical structure of main active constituents in herbal extracts.

Source	Active molecule	Structure	Functions
*Abronia nana* S.Watson	Boeravinone X	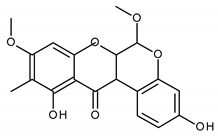	Antiseptic, inhibits HMGB1 release and HMGB1-mediated hyperpermeability leukocyte adhesion, migration, and cell adhesion molecule expression.
*Aconitum carmichaelii* Debeaux	C19-diterpenoid	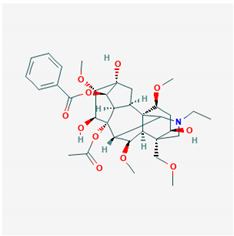	Inhibition of TLR4, NF-κB, HMGB1, and caspase-3 expression in injury.
*Angelica sinensis* (Oliv.) Diels	Ferulic Acid	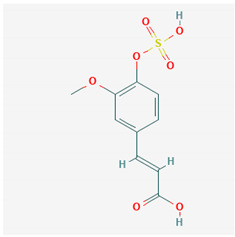	Anti-inflammatory. No effect on HMGB1. HIV protease inhibition.
	Angelica sinensis polysaccharide	N/A	Viral replication inhibition, anti-inflammatory.
Pelargonidin	Pelargonidin	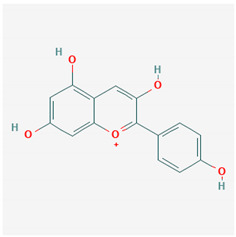	Inhibits HMGB1 release and HMGB1-mediated hyperpermeability leukocyte adhesion, migration, and cell adhesion molecule expression. Inhibits HMGB1 mediated production of TNF-α, IL-6, and activation of NF-κB and ERK1/2.
*Aspalathus linearis* (Burm.f.) R.Dahlrgen (Rooibos)	Aspalathin	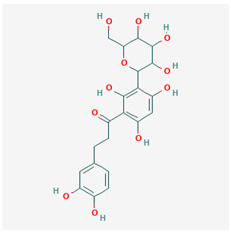	Inhibits HMGB1 release and HMGB1-mediated hyperpermeability leukocyte adhesion, migration, and cell adhesion molecule expression.
	Nothofagin	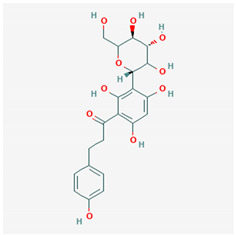	
*Astragalus mongholicus* Bunge	Astragalus mongolicus polysaccharide	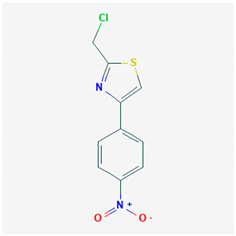	Inhibits HMGB1-induced hyperpermeability of ECs.
Cucurbitaceae sp.	Cucurbitacin E	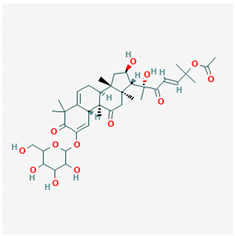	Ameliorate EC insult by blocking HMGB1-TLR4-NF-κB.
*Cyclopia intermedia* E.Mey. (Honeybush)	Vicenin-2	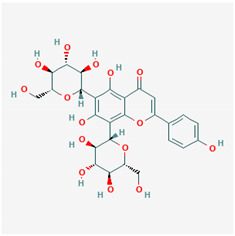	Inhibits HMGB1 release and HMGB1-mediated hyperpermeability leukocyte adhesion, migration, and cell adhesion molecule expression. Inhibits HMGB1 mediated production of TNF-α, IL-6, and activation of NF-κB and ERK1/2.
	Scolymoside	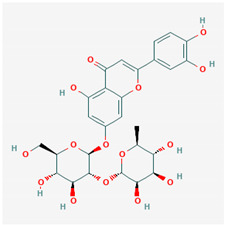	
Ecklonia cava	Dieckol	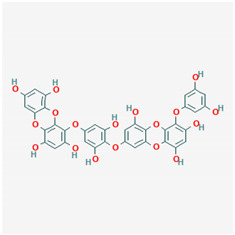	Attenuates serum levels of NO, PGE2, and HMGB-1. Down-regulates macrophage levels of iNOS, COX-2, TNF-α, IL-6, and HMGB-1. Inhibits SARS-CoV 3CL^pro^.
*Flavonoides* (various)	Luteolin	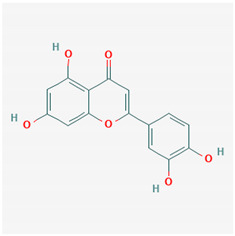	Reduces HMGB1 release. Inhibits SARS-CoV 3CL^pro^.
	Quercetin	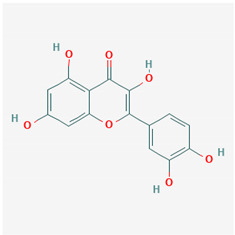	Reduces lung permeability, COX-2, HMGB1, iNOS expression, and NF-κB p65 phosphorylation. Inhibits SARS and MERS-CoV 3CL^pro^.
	Baicalein	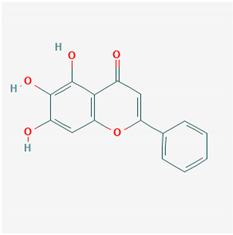	Down-regulates MMP 2 and 9. Attenuates HMGB1 translocation from nucleus to cytoplasm. Inhibits SARS-CoV 3CL^pro^.
	Resveratrol	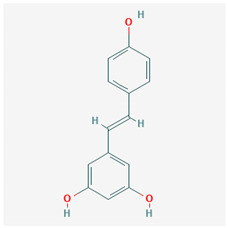	Induces Sirt1 which leads to HMGB1 nuclear retention.
*Panax ginseng* C.A.Mey.	Ginsenoside R1	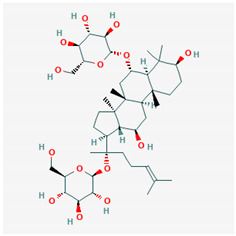	Reduces HMGB1 release. Inhibits NF-κB activation thus reducing TNF, IL-1, IL-6 and IFN-γ production. Inhibits hemagglutinin.
*Camellia sinensis* (L.) Kuntze	Epigallocatechin Gallate	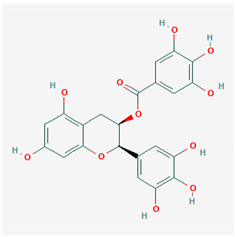	Inhibits neuraminidase and hemagglutinin activity. Reduces HMGB1 release. Prevents HMGB1 accumulation on macrophage cell surface.
*Inula helenium* L.	Alantolactone	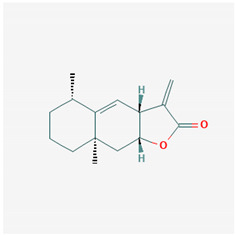	Inhibits NF-κB, IκBα. Suppresses IL-8, TNF-α and IL-1β, ICAM-1 and VCAM-1 release.
*Actinidia argute* (Siebold and Zucc) Planch ex Miq. (Kiwifruit)	Procyanidin	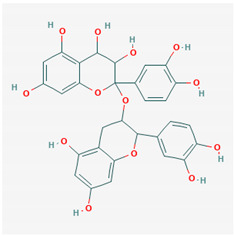	Inhibits production of IL-6, IL-1β, TNF-α pro-inflammatory cytokines, HMGB1 and granzyme B serine protease by activated monocytes.
*Glycyrrhiza glabra* L. (Licorice)	Glycyrrhizic acid	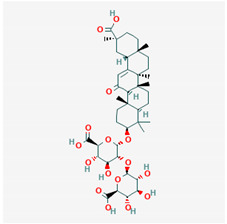	Decreases TNF-α, IL-1β, and HMGB1 production.
	Glycyrrhetinic acid	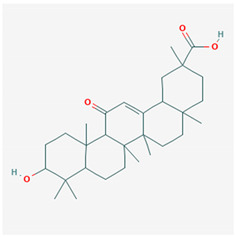	Stimulates Sirt6 expression, which leads to HMGB1 nuclear retention. Blocks extracellular HMGB1. Directly binds HMGB1.
*Lonicera caprifolium* L.	Chlorogenic acid	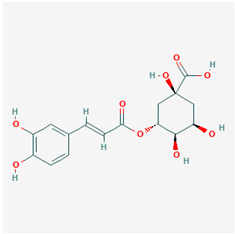	Decreases TNF-α, IL-1β, and HMGB1 levels in blood.
*Perilla frutescens* (L.) Britton	Rosmarinic acid	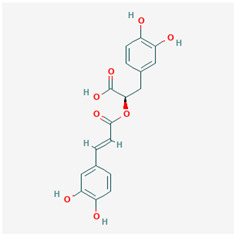	Down-regulates HMGB1 inflammatory response. Inhibits HMGB1-mediatedhyperpermeability and leukocyte migration.
*Plumbago zeylanica*	Plumbagin	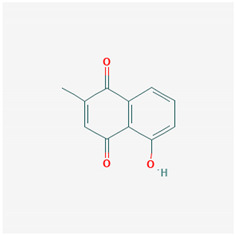	Inhibits HMGB1 expression, NF-κB, TNF-α, MPO activity.
*Prunella vulgaris* L.	Caffeic acid	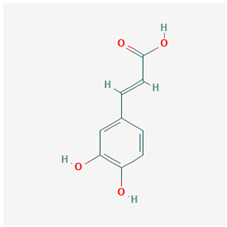	Inhibits HMGB1 release, serum levels and expression.
*Rhodiola rosea* L.	Salidroside	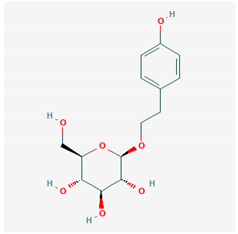	Inhibits HMGB1 expression. Stimulates Sirt1 expression. Decreases serum TNF-α, IL-6, NO, HMGB1 levels and iNOS, NF-κB-p65 expression.
*Salvia miltiorrhiza* Bunge	Danshensu	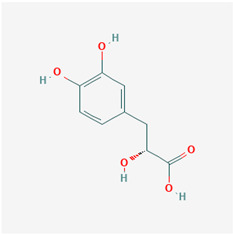	Inhibits HMGB1 levels. Antiviral, hepatitis B virus reverse transcriptase inhibition.

Majority of the molecules presented above belong to the polyphenol family: rotenoids, diterpenes, phenolic acids, flavonoids, phlorotannin, triterpene saponins, stilbenes and phenylpropanoids. An exception is the Sesquiterpene lactone, which has an active lactone ring with an exomethylene group, and a polysaccharide composed of simple sugars with hydroxyl groups which give the molecule polarity and an anti-inflammatory activity. Images reproduced from PubChem database [156].

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
