# Peer review of "Influence of Herbal Medicines on HMGB1 Release, SARS-CoV-2 Viral Attachment, Acute Respiratory Failure, and Sepsis. A Literature Review"

_ijms, 2020, doi:10.3390/ijms21134639_

Round 1

Reviewer 1 Report

In this review, the authors reported published anti-HMGB1 and antiviral activities of herbal extracts in order to suggest new potential therapeutic strategies against SARS-CoV-2 and/or COVID-19.

There are some criticisms that should be addressed/considered.

Specific points are raised below.

Major points

  1. The title is misleading. The review focuses also on the antiviral activity of herbal medicines, not only on anti-HMGB1 properties.
  2. One of the main critical points concerns the data on ACE and ACE-2. The authors sometimes report confusing information about the roles of the two molecules, as if they were interchangeable. ACE and ACE-2 are two main actors, often involved in the same pathways but with conflicting activities. Other activities are peculiar to one but not to the other.
  3. The authors report antiviral activities of numerous herbal medicines, but they do not consider the specificity for particular viruses (e.g. different host receptors or proteases involved). This should be taken into consideration when the authors aim to suggest a therapeutic potential against SARS-CoV-2.
  4. The authors report a long list of herbal extracts/medicines describing their potential anti-HMGB1 activity. Many of these medicines have also anti-inflammatory properties. As the authors describe in the discussion, many of these data do not clarify if there is a direct anti-HMGB1 activity, or if these compounds decrease HMGB1 release by reducing inflammatory processes, ultimately affecting HMGB1, in accordance with the timing of the effects on HMGB1. If a clear interaction (e.g. binding) between HMGB1 and a herbal extract has been demonstrated, authors should indicate it.
  5. It would be useful to organize the review with one paragraph on the impact of herbal medicines on viral infection and one on the effect on HMGB1. If data on direct impact are available, I would suggest including it in the main text and perhaps those with an uncertain direct impact on Table 1 (with bibliographic references included).
  6. The paper should be carefully edited by an English native speaker.

Author Response

n.b. all line numbers are correct when the “track changes” function is turned on.

Point 1: The title is misleading. The review focuses also on the antiviral activity of herbal medicines, not only on anti-HMGB1 properties.

Response 1: Both reviewers identified the same issue. Title has been amended to provide a reference to antiviral properties outlined in the herbal medicine review. New title now: “Influence of herbal medicines on HMGB1 release, SARS-CoV-2 viral attachment, acute respiratory failure, and sepsis. A literature review.” A broader title omitting SARS-CoV-2 would be improper as the review did not exhaustively research the selected herbal extracts across viral attachment inhibition of viruses other than SARS-CoV-2, and has only made fleeting references to other viruses where these could serve to support the broad proteolytic effect of the extracts as beneficial against HMGB1 release, acute respiratory failure and/or sepsis.

Point 2: One of the main critical points concerns the data on ACE and ACE-2. The authors sometimes report confusing information about the roles of the two molecules, as if they were interchangeable. ACE and ACE-2 are two main actors, often involved in the same pathways but with conflicting activities. Other activities are peculiar to one but not to the other.

Response 2: The text does not intend to draw a parallell between ACE and ACE-2, other than highlighting their homology, proteolytic activity and therefore a similar inhibitory effect of herbal medicines – even though their structure and functinon are different and even physiologically opposite. In the introduction there is a general explanation of which physiological systems are modulated by the ACE-2 enzyme.  Chapter beginning at line 95 is now edited with more specific details of ACE-2 physiology. In subheading 2.1 there are three sentences mentioning ACE in contradistinction to ACE-2. Line 164 has been amended to remind the reader of their opposing function. Line 174 highlights a lack of evidence with regards to ACE-2 in as compared to ACE. Line 197 ACE was corrected to state ACE-2.

Point 3: The authors report antiviral activities of numerous herbal medicines, but they do not consider the specificity for particular viruses (e.g. different host receptors or proteases involved). This should be taken into consideration when the authors aim to suggest a therapeutic potential against SARS-CoV-2.

Response 3: Care has been taken to remove broad generalisations from the descriptions of each herbal medicine regarding their speculative antiviral properties, and instead emphasis has been put on the virus which was in fact tested for, and the host receptor or protease if present. look line 269, 283, 325, 341, 370, 407, 417, 509.

In the discussion it has however been maintained that a non-specific protease-inhibitory function of phenolic extracts could have a broad activity on slowing down viral infections, due to the homologies between proteases and the broad inhibitory functioning of phenolic compounds, which also serves to prevent the activation of HMGB1. Therefore references to non-SARS-CoV-2 viruses have been retained in the main body. As in-depth analysis of various viral attachment proteins is not the aim of this article, a table summarising virus – host receptor – protease involved has not been created, and the authors expect that just as with abbreviations, the first mention will be considered authoritative, and the reader can infer the connections at each subsequent mention. However, to reiterate, the aim of the article is to indicate that a proteolytic activity is employed by a broad range of viruses to gain entry in to the cell. That a broad enzyme – inhibitory activity is evident across a range of herbal extracts. And that many of these extracts happen to have a common chemical structure which can be grossly classified as phenolic, and that this structure can also be correlated to the anti-inflammatory, HMGB1 inhibitory effect and to the anti-viral effect. These points if achieved would hopefully encourage the reader to research a clinical application. It is worthwhile to note that the authors also do not specifically suggest phenolic SARS-CoV-2 therapeutic potential, and instead focused the discussion on treating the symptom which accompanies a SARS-CoV-2 disease process, which is shared across other viral infections, and that is a septic state which is the major cause of death.

In order to clarify this conclusion, a statement at line 570 has been added to highlight the broad antiviral function of the selected herbal compounds. And hopefully the conclusion is adequate to cement the idea at line 614

Point 4: The authors report a long list of herbal extracts/medicines describing their potential anti-HMGB1 activity. Many of these medicines have also anti-inflammatory properties. As the authors describe in the discussion, many of these data do not clarify if there is a direct anti-HMGB1 activity, or if these compounds decrease HMGB1 release by reducing inflammatory processes, ultimately affecting HMGB1, in accordance with the timing of the effects on HMGB1. If a clear interaction (e.g. binding) between HMGB1 and a herbal extract has been demonstrated, authors should indicate it.

Response 4: It is hard to decide whether inhibition of release of HMGB1 should be considered a direct anit-HMGB1 activity, since it is a sufficient effect on HMGB1 to support our argument. In a similar vein it is difficult to establish whether HMGB1 receptor binding (resulting in competitive inhibition) should also be considered direct anti-HMGB1 activity. Reference papers have been combed for all modes of interaction of the herbal medicines with HMGB1, any of which – if relevant – were then mentioned in the description of each herbal extract. However, the data has been re-evaluated to find if any signs of binding to HMGB1 were not parapharsed in the main body of the article. This has then been corrected (see line 378, 389 ) and updated in the table 1 column “functions” to include “Direct binding of HMGB1” or “via direct binding”.

A new column which would distinguish extracts with a direct HMGB1 binding was considered confusing, as for the purposes of mitigating HMGB1 mediated sepsis direct HMGB1 binding does not necessarily result in a better outcome than inhibition of gene transcription, nuclear or cytoplasmic release. It would also be insufficiently helpful as too few of the herbal extracts have been assayed for binding to the HMGB1 molecule. A suggestion for further research in this matter is present at line 645

Point 5: It would be useful to organize the review with one paragraph on the impact of herbal medicines on viral infection and one on the effect on HMGB1. If data on direct impact are available, I would suggest including it in the main text and perhaps those with an uncertain direct impact on Table 1 (with bibliographic references included).

Response 5: As in point 4, it would be counter to the aim of the review if extracts which had no direct binding to HMGB1 were completely redacted and left as a side note in the table. Not only due to the HMGB1 binding but also due to the anti-viral properties listed and the broader look at the signalling pathways. Since there is no clear cut as to which herbal extracts have more breadth on the topic of viral attenuation or HMGB1, to avoid confusion the full descriptions are left in the text.

Since a majority of the studies reviewed did not focus on direct attachment of the herbal extracts to HMGB1, this gap has been highlighted in the conclusion as a promising future endeavour. See line 510 and 645.

Structurally the review attempts to follow from most contemporary approaches in antiviral therapies – targeting viral binding and transcription molecules – to suggesting an alternative symptomatic management with herbal medicines. It is therefore on purpose that the body begins with ACE-2, follows on to HMGB1, and finishes with a variety of herbal extracts with multiple modes of action both on ACE-2 and HMGB1 as well as non-SARS-COV-2 viral activity. This hopefully serves to broaden the definition of clinical virus management. Also focusing in on the breadth of targets not classically considered in virus-host or cytokine-receptor attachment, namely the glycans. Therefore, adjusting the structure to fit a specific scheme of information re-organisation is counterproductive if that theme is not in line with the main argument of the review, and also can be left as an exercise to the reader as all the data found in the review is identified in the description of each ehrbal extract within the main body.

Point 6: The paper should be carefully edited by an English native speaker.

Response 6: The paper has been edited for grammatical clarity with numerous edits present throughout the manuscript, hopefully the text as a whole is now more easily accessible.

Reviewer 2 Report

I want to thanks the editorial board for giving me the opportunity to review this article

In this review, Wyganowska-Swiatkowska et al.  aim to describe several therapeutic options (herbal medicines) to modulate or fight virus infection.

When reading the introduction, the subject is undoubtedly interesting: is it possible, through other therapeutic approach while waiting for vaccine availability to limit the sepsis like immune reaction of virus infection? Unfortunately, the manuscript does not have the proper structure to answer that particular question. Furthermore, it is not clear whether the authors want to study sars-cov2 or sepsis induced respiratory failure.

I suggest that the authors rewrite the introduction, which is the most important part of the manuscript suffers from lack of references and some sentences are incorrect.

For the introduction, I would rather follow:

  1. Sars-Cov2 related acute respiratory failure
  2. Sars-Cov2 induced sepsis like syndrome
  3. HMGB1 as a possible mediator in SARS-COV2 (although there is no data): data on HMGB1 and influenza or Sars-Cov 1.

Then, in the section interventions, I would rather use the following part:

  1. Suppress the paragraph on Sars-Cov2 and its connection with ACE-2
  2. Effects of HMGB1 inhibition on virus-induced respiratory failure and sepsis-like syndrome (separate animal model and human studies)
  3. Influence of herbal medicines on HMGB1/respiratory failure
  4. Use a figure to illustrate

Other comments:

  1. Article title does not reflect the aim of the article: virus related respiratory failure should be highlighted.
  2. The term acute respiratory system breakdown is not relevant. The authors should use ARDS or acute respiratory failure.
  3. Page 2, line 94: “this is the beginning of sepsis”. Sepsis has a rigorous definition
  4. The manuscript suffers from a lack of references and some sentences are incorrect

Author Response

n.b. all line numbers are correct when the “track changes” function is turned on.

Point 1: When reading the introduction, the subject is undoubtedly interesting: is it possible, through other therapeutic approach while waiting for vaccine availability to limit the sepsis like immune reaction of virus infection? Unfortunately, the manuscript does not have the proper structure to answer that particular question. Furthermore, it is not clear whether the authors want to study sars-cov2 or sepsis induced respiratory failure.

I suggest that the authors rewrite the introduction, which is the most important part of the manuscript suffers from lack of references and some sentences are incorrect.

For the introduction, I would rather follow:

  1. a) Sars-Cov2 related acute respiratory failure
  2. b) Sars-Cov2 induced sepsis like syndrome
  3. c) HMGB1 as a possible mediator in SARS-COV2 (although there is no data): data on HMGB1 and influenza or Sars-Cov 1.

Response 1: While it is true that the review can not answer whether it is possible to limit the sepsis response during a late stage SARS-CoV-2 infection, it aims to suggest that there is a possibility which has not yet been investigated in human trials. Therefore the structure of the introduction is as follows: introduction of viruses, distinction between RNA and DNA viruses   - and a focus on RNA viruses as the more genetically variable. An explanation as to why this can cause problems with finding a vaccine or a reliable therapeutic agent. Examples of these difficulties in other SARS viruses. The similarity between previous SARS infections and the current one. Similarities between the general mechanisms of infection between viruses. Explanation of some of the underlying cell signalling pathways which allow for this attachment, as well as their inflammatory response. There is a section missing on the importance of glycan recognition in RAGE binding as well as SARS S protein binding to TMPRSS or influenza hemagglutinin bindig to HAT, and finally modified SARS S protein binding to ACE-2. The last part of the introduction suggests that herbal medicines have been found to play a part in inhibiting all of these glycan – sensitive attachment mechanisms. The importance of glycan binding is then picked up again after all the data regarding the selected herbal medicines is presented, in the discussion.

In order to fix the introduction, a missing section has been added at line 142 to illustrate how important glycans are in the identified processes of HMGB1-RAGE bnding, SARS S protein – TMPRSS binding, and SARS S protein – ACE2 binding.

An explanation why this glycan binding is important with regard to herbal extracts has been added at the end of the introduction at line 150

references have been added to line 45, 49, 51, 54, 55, 61, 64, 67, 69, 73, 84, 138, 143 and 145 and confusing sentences redacted at line 73, 75 and 95

Point 2: Then, in the section interventions, I would rather use the following part:

  1. a) Suppress the paragraph on Sars-Cov2 and its connection with ACE-2
  2. b) Effects of HMGB1 inhibition on virus-induced respiratory failure and sepsis-like syndrome (separate animal model and human studies)
  3. c) Influence of herbal medicines on HMGB1/respiratory failure
  4. d) Use a figure to illustrate

Response 2: a) ACE-2 is an important part of the SARS-CoV-2 infection pathway, and it illuminates the importance of proteolytic inhibition which plays a key role in the discussion and was therefore thought to be vital in its entirety for the text as it is. It has been edited for clarity regarding the difference between ACE and ACE-2.

  1. b) due to the scarcity of human studies it was decided that a separation between human and animal studies would not add sufficient clarity. The conclusion encourages more human studies to be carried out in line 647.
  2. c) The herbal medicines section already covers HMGB1 and respiratory failure influences.
  3. d) Table 1 illustrates the rich variety in superstructure of the molecules while also highlighting the common moieties and functions, which serve to support the main argument of the review.

Point 3: Article title does not reflect the aim of the article: virus related respiratory failure should be highlighted.

Response 3: Both reviewers identified the same issue. Title has been amended to provide a reference to antiviral properties outlined in the herbal medicine review. New title now: “Influence of herbal medicines on HMGB1 release, SARS-CoV-2 viral attachment, acute respiratory failure, and sepsis. A literature review.”. A broader title omitting SARS-CoV-2 would be improper as the review did not exhaustively research the selected herbal extracts across viral attachment inhibition of viruses other than SARS-CoV-2 and has only made fleeting references to other viruses where these could serve to support the broad proteolytic effect of the extracts as beneficial against HMGB1 release, acute respiratory failure and/or sepsis.

Point 4: The term acute respiratory system breakdown is not relevant. The authors should use ARDS or acute respiratory failure.

Response 4: Terminology reviewed to replace all mentions of acute respiratory system breakdown with acute respiratory failure.

Point 5: Page 2, line 94: “this is the beginning of sepsis”. Sepsis has a rigorous definition 

Response 5: Previously line 94, now line 136, has been amended to clarify and expand on evidence for defining the state described in the paragraph as septic, with reference to a defintion. 

Point 6: The manuscript suffers from a lack of references and some sentences are incorrect

Response 6: incorrect sentences and missing references amended see point 1 for introduction, but also look line 164, 180, 205, 224, 231, 282, 326, 390, 509, 564, 570, and line 627. other small changes done also during a re-editing for correctness of English.

Round 2

Reviewer 1 Report

The review has greatly improved. Only minor spell check required.

Author Response

First of all, we would like to thank you for your time, professional review and amiability to our paper.

Point 1: The review has greatly improved. Only minor spell check required.

Response 1: Thank you. A spell check has been carried out, with multiple corrections throughout the text.

Reviewer 2 Report

The manuscript has been improved according to the suggestions of reviewers. However, I suggest that the authors include recent data from COVID-19 associated immune dysfunction demonstrating that patients with COVID-19-associated ARDS usually met the diagnosis criteria for sepsis-associated immunosuppression (Tay MZ, Poh CM, Renia L, MacAry PA, Ng LFP: The trinity of COVID-19: immunity, inflammation and intervention. Nat Rev Immunol 2020), including enhanced susceptibility to develop nosocomial infections. 

Furthermore, line 74 "massive neutrophil infiltration and acute lung injury".First reports found that interstitial mononuclear inflammatory infiltrates were dominated by lymphocytes. This sentence needs to be corrected

Author Response

Point 1: The manuscript has been improved according to the suggestions of reviewers. However, I suggest that the authors include recent data from COVID-19 associated immune dysfunction demonstrating that patients with COVID-19-associated ARDS usually met the diagnosis criteria for sepsis-associated immunosuppression (Tay MZ, Poh CM, Renia L, MacAry PA, Ng LFP: The trinity of COVID-19: immunity, inflammation and intervention. Nat Rev Immunol 2020), including enhanced susceptibility to develop nosocomial infections.

Response 1: Thank you. Line 110 has been added to illustrate the current progress in using immunosuppressive and anti-inflammatory drugs in the treatment of COVID-19 and prevention of associated lung damage with reference to (Tay MZ et al. 2020), focusing on the limitations in the current success of trials. A note on the importance of appropriately controlling nosocomial infections along with improving patient survivability has been added in the same section with appropriate references.

Point 2: Furthermore, line 74 "massive neutrophil infiltration and acute lung injury".First reports found that interstitial mononuclear inflammatory infiltrates were dominated by lymphocytes. This sentence needs to be corrected

Response 2: Previously line 74, now line 80, has been amended and now reads “a massive lymphocyte infiltration associated with acute lung injury” with supporting references.